# Abductive Reasoning with Probabilistic Commonsense

**Joseph Cotnareanu** [1 2 3] **Chiara Roverato** [1 2 3] **Han Zhou** [4] **Didier Chetelat** [4] **Yingxue Zhang** [4] **Mark Coates** [1 2 3]

## Abstract

Recent efforts to improve the reasoning abilities of Large Language Models (LLMs) have focused on integrating formal logic solvers within neurosymbolic frameworks. A key challenge is that formal solvers lack commonsense world knowledge, preventing them from making reasoning steps that humans find obvious. Prior methods address this by using LLMs to supply missing commonsense assumptions, but these approaches implicitly assume universal agreement on such commonsense facts. In reality, commonsense beliefs vary across individuals. We propose a probabilistic framework for abductive commonsense reasoning that explicitly models this variation, aiming to determine whether most people would judge a statement as true or false. We introduce Probabilistic Abductive CommonSense (PACS), a novel algorithm that uses an LLM and a formal solver to sample proofs as observations of individuals' distinct commonsense beliefs, and aggregates conclusions across these samples. Empirically, PACS outperforms chain-of-thought reasoning, prior neurosymbolic methods, and search-based approaches across multiple benchmarks.

## 1. Introduction

Much research in recent years has been centered on improving the abilities of Large Language Models (LLMs) to reason formally. One particularly promising approach revolves around providing the LLM access to a formal logic solver (Ye et al., 2023; Lyu et al., 2023; Olausson et al., 2023; Lee and Hwang, 2024). Fueled by LLMs' strong

abilities to translate between natural language and formalized logic (Yang et al., 2024), this revival of neurosymbolic approaches in AI has the potential to finally lead to human-level ability on reasoning benchmarks.

In the simplest possible approach, the LLM translates the reasoning problem into a set of formal logic premises $S$ and a query $c$, and a solver is called to determine whether $S$ entails $c$ or $\neg c$ (Ye et al., 2023; Yang et al., 2024). However, this naive approach is often brittle, as solvers are often incapable of solving the problem because they lack access to necessary commonsense information, such as "white is a color" or "rabbits are animals". LLM-Tres and ARGOS (Toroghi et al., 2024; Cotnareanu et al., 2026) attempt to solve this problem by using the LLM to build an additional set of commonsense propositions $L$ that complete the problem, in the sense that anybody would agree $L$ is true ($\vdash L$) and $S \wedge L \vdash c$ or $S \wedge L \vdash \neg c$. This turns the problem from deductive to abductive reasoning, where not only the truth value of $c$ must be found, but also the missing information $L$ necessary to pin down this value.

While this approach is very valuable in some settings where the concepts in question are uncontroversial (e.g., "hot sauce is spicy"), it often fails in more complex or ambiguous domains. Indeed, when a question might involve more fluid topics, there might simply not be any single set of additional facts $L$ that completes the problem and that everybody would agree is true ($\vdash L$). In these domains, two people might disagree about the truth value of a query $c$ not because of errors in reasoning, but rather because one holds a set of commonsense beliefs $L_1$ such that $S \wedge L_1 \vdash c$, while the other holds a different set of commonsense beliefs $L_2$ such that $S \wedge L_2 \vdash \neg c$, and those belief sets are both defensible yet incompatible with each other.

Another clear down-side of assuming a monolithic commonsense is that expressing clearly a probability of the truth of a proposition becomes difficult. In LLM-Tres, for example, a probability of entailment between two variables is invoked frequently, and relied upon heavily by the method. However, since the source of the randomness of the truth is never described, this probability is not interpretable or understandable and is in truth more of a confidence-score.

[1]McGill University [2]International Laboratory on Learning Systems [3]Mila - Quebec Artificial Intelligence Institute [4]Huawei Noah's Ark Lab. Correspondence to: Joseph Cotnareanu <joseph.cotnareanu@mail.mcgill.ca>.

*Proceedings of the 43rd International Conference on Machine Learning*, Seoul, South Korea. PMLR 306, 2026. Copyright 2026 by the author(s).

In this work, we propose a novel neurosymbolic and probabilistic framework for abductive reasoning problems which utilizes both the strengths of LLMs and formal logic solvers, but without the problematic assumption that everybody would agree on the missing assumptions. Instead, we propose to define the "abductive probability" of a query being true as its average over the commonsense set $L(R)$ of every human being $R \in \text{Humans}$:

$$\text{AP}(S, c) \equiv \text{E}_R\big[\mathbb{1}[S \wedge L(R) \vdash c]\big],$$

and say the query $c$ is "abductively true" given the premises $S$ if more people would conclude it is true than false. Then, we propose to derive an algorithm that aims to determine whether a query $c$ is abductively true or false, so that it can be presented to the user as the "correct" answer.

In detail, we devise a sampling algorithm that uses a large language model and a logic solver to produce plausible commonsense vectors $L_1, L_2, \ldots, L_K$ which indicate personal belief. This algorithm involves sampling successive propositions, and prioritizes reaching a conclusion ($S \wedge L_k \vdash c$ or $S \wedge L_k \vdash \neg c$) as fast as possible, so as to minimize computational cost. Given these sampled commonsenses, we then approximate the probability of a query being true or false by a Monte-Carlo estimate,

$$\widehat{\text{AP}}(S, c) = \frac{1}{K} \sum_{k=1}^{K} \mathbb{1}[S \wedge L_k \vdash c],$$

and conclude the query being true or false based on whether this estimate is greater or smaller than 0.5.

On various abductive reasoning benchmarks, we show that this approach, which we call Probabilistic Abductive CommonSense (PACS)[1], improves accuracy compared to both chain-of-thought (Wei et al., 2022) and prior neurosymbolic approaches, demonstrating its value as a reasoning algorithm as well as the credibility and value of the underlying assumptions and framework used for its derivation.

## 2. Related Work

Our work is closely related to the prior literature on using LLMs to solve reasoning problems, both with and without logic solvers. In addition, our sampling algorithm is closely related to methods that employ an LLM to search for optimal reasoning steps in a tree-like fashion.

**Neural Methods** Wei et al. (2022) presented the first framework for LLM-based reasoning. In the work, it was shown that providing examples of rationales for answers to questions can induce the LLM to do the same, leading to improved accuracy. Kojima et al. (2022) demonstrated that

---

[1]Get our code on GitHub.

a simple prompt can have a similar effect as providing examples of reasoning to induce such reasoning: prepending the sentence "Let's think step by step" before generating an answer. This is known as "Chain of Thought" (COT). Following this, Wang et al. (2023) proposed self-consistency (SC), using COT multiple times and taking the vote as the prediction. However, Saparov and He (2023) observed that COT and SC suffer from challenges in proof planning — steps in the COT tend to be factual but do not always actually contribute to the answering of the problem. This motivates guidance of the LLM at a step-level. Kazemi et al. (2023) and Lee and Hwang (2024) proposed more logic-focused methods, with reverse reasoning, starting at the answer and ending at the problem. These back-chaining methods, however, underperform symbolic approaches.

**Symbolic Methods** Considering that LLMs are poor proof-planners, a series of methods, including F-COT (Lyu et al., 2023) and SAT-LM (Ye et al., 2023), proposed to execute the reasoning with more specialized, logical tools. In these works, the LLM converts the text to symbolic logic, and a solver is then employed. Logic-LM (Pan et al., 2023) extended this to include a self-refinement step. While these methods perform well on simple or extremely logically structured datasets, they fail to account for ambiguity and the exclusion of common knowledge. Addressing this, Liu et al. (2024) and Wang et al. (2022) proposed algorithms that produce new clauses via logical deduction and then augment the COT prompt with these mined logical relations. While this might help the LLM, it does not add information to the problem, because any added relations are already deducible (since they are generated purely via logical manipulation). Instead of producing clauses via deduction, Toroghi et al. (2024) proposed a method that exhaustively searches for new single-proposition modus-ponens clauses. However, the search is conducted only over the propositions from the question, and repeated until the problem is solvable by classical logic, diminishing robustness. This search space is highly restricted and leaves out nearly all necessary information for some logic problems. To address this, Cotnareanu et al. (2026) propose a similar method of abductive clause search, replacing the exhaustive methodology over a restricted clausal space from LLM-Tres with a greedy, LLM-driven search over the comprehensive space of all possible clauses.

**Tree Search Methods** Yao et al. (2023) proposed Tree of Thoughts (TOT), which explores hand-crafted trees using an LLM to solve reasoning tasks. TOT is poorly suited to logical reasoning settings as logic problems have highly variable tree-structures. Hao et al. (2023) likened LLM reasoning problems to planning problems given a world model. The authors then applied a standard planning solution, Monte Carlo Tree Search (MCTS), to LLM reasoning.

The authors make use of sampled confidence scores (the frequency of the same step being sampled over many samples), as well as token-level confidence scores, as search heuristics. MCTS requires simulation in order to estimate future rewards, which means that at each step, enough complete reasoning paths must be generated from each candidate step so that the empirical expectation of the future reward is meaningful. This leads MCTS to be extremely expensive computationally.

The computational burden of MCTS led to interest in cheaper alternatives, such as beam search. Xie et al. (2023) proposed a step-wise beam-search in which the LLM itself evaluates the score of each candidate step. The need for LLM-based scoring means that either a small number of candidate thoughts must be considered over the course of the search, or that we must incur significant overhead expense, since each candidate step generated requires a generative, LLM-based analysis and scoring.

Wang et al. (2025) propose a beam-search algorithm which uses a two-fold score: both step-novelty and step-informativeness are measured. A step is considered novel if its conclusion is not frequently referenced in the chain, and it is considered informative if the conclusion is not frequently mentioned in other candidate thoughts at the same step. This method relies upon trigram similarity, making it non-robust to semantics-preserving linguistic variation. It is also heavily dependent on programmatic text-parsing in order to identify the conclusion of a step.

## 3. Problem Statement

In this section, we outline formally the problem of abductive commonsense reasoning. We define a logic problem $(S, c)$ as a pair consisting of logical context $S$ and a specific True/False question $c$ (defined more formally below). The problem is abductive if the information in $S$ is insufficient to know $c$. We define a random variable $L$ which is random over "reasoners", who are most easily understood as individual people; each realization $L$ indicates all of an individual's beliefs so that we can now, using both $S$ and $L$, determine whether $c$ is True or False.

We introduce the nuance that we do not have access to complete belief-vectors $L$, only the ability to sample beliefs on individual elements of $L$ (i.e., "thoughts" in a COT sense). We are interested in minimizing the number of "thoughts" which we must access to determine the truth of $c$.

### 3.1. Preliminaries

Let $\mathcal{D}$ be an ordered set of all English sentences. Each proposition $D_i \in \mathcal{D}$ describes some concept (where $D_i$ is the $i$-th element in $\mathcal{D}$), for which there are three associated epistemic statements: $e_i \in \mathcal{E} = \{$I know, I know that

not, I don't know whether or not$\}$. In this way, $e_i$ tells us whether proposition $D_i$ is True, False, or unknown. Let $\mathcal{P}(\mathcal{D})$ denote the space $\mathcal{E}^{|\mathcal{D}|}$. Let $c \in \mathcal{D}$ be a logical proposition whose truth we want to evaluate, given some logical context $S \in \mathcal{P}(\mathcal{D})$ such that $S$ is insufficient to deduce logically whether $c$ is True (or false); i.e. $(S \not\vdash c) \wedge (S \not\vdash \neg c)$.

For $c \in \mathcal{D}$ and $\mathcal{Z} \in \mathcal{P}(\mathcal{D})$ we introduce a mapping $T : \{\mathcal{D}, \mathcal{P}(D)\} \rightarrow \{True, False, Unknown\}$. This is binary logical truth function of the proposition represented by $c$ given the context $\mathcal{Z}$:

$$T(c, \mathcal{Z}) = \begin{cases} True, & \text{if } \mathcal{Z} \vdash c, \\ False, & \text{if } \mathcal{Z} \vdash \neg c, \\ Unknown, & \text{if } (\mathcal{Z} \not\vdash c) \wedge (\mathcal{Z} \not\vdash \neg c). \end{cases} \quad (1)$$

For our considered pair $(c, S)$, since $S$ is insufficient to know whether $c$ is True ($T(c, \mathcal{S}) = Unknown$), we understand that additional information must be added during the reasoning process.

We introduce a random variable $L = g(R)$. Here $R \in \mathcal{R}_{S,c}$ is a random reasoner selected according to a probability distribution $p(R)$ from the set of reasoners $\mathcal{R}_{S,c}$ who are qualified to solve the problem $(S, c)$. The function $g$ maps from the selected reasoner to $L$, which is a vector of epistemic statements, each in $\mathcal{E}$, representing all of the reasoner's beliefs over $\mathcal{D}$. We assume that all reasoners are logically consistent (i.e., they hold no two contradictory beliefs). We require that each reasoner know sufficient information for the truth of $c$ to be determined (from the perspective of that reasoner), so that $L \in \Omega_{S,c} \subset \mathcal{P}(\mathcal{D})$ where $\Omega_{S,c} = \{\mathcal{X} : \mathcal{X} \subset \mathcal{P}(\mathcal{D}), (\mathcal{X} \wedge S \vdash c) \vee (\mathcal{X} \wedge S \vdash \neg c)\}$. Let $L_i$ represent the $i$-th element in the vector $L$, so that it specifies an epistemic value on the concept $D_i$.

Under this vector representation of epistemic knowledge, for a realization $l$ of $L$, logical operations, e.g., $S \wedge l$, require the conversion of the vector form to logical form, so that the logical form of $l$ is $\wedge_{i=1}^{|D|} l_i$, where $l_i$ is the epistemic statement regarding a specific proposition $D_i$. Throughout the text, whenever we write logical expressions using vectors, the conversion to logic is left implicit but always done in the above fashion. The probability distribution over reasoners $p(R)$ thus induces a probability distribution $p_{S,c}(L)$. For most problems, the full descriptions of all beliefs, $l$, is not necessary. Only a few relevant concepts in $\mathcal{D}$ are required. So, a major element of the problem is finding ways to sample entailment over reasoners without requiring a full sample $l$. We assume that there is only access to a sampler of individual epistemic statements $l_i$, with a constant, known cost associated with drawing a sample. The goal of this problem setting is twofold:

1. Accurately determine the truth of $c$ (where truth is defined by an expectation over the reasoners).

2. Do so at minimal cost.

# 4. Method

Here, we will describe our method and its derivation. To lend the following section some concreteness, we provide the an example in Figure 1 in which a simple question is asked about whether a gift should be brought for a birthday party in which no gift-related instructions were given. Clearly, this is an ambiguous problem which can only be decided by applying one's personal sense. In this example, we can see several potential seemingly-reasonable and yet contradictory ways to answer the question. The method which we propose below aims to search, over the tree of possible reasoning steps, for diverse and potentially contradictory paths so that each path answers the question clearly and differently. In short, we use a **model count** on the logical form of the problem as a proxy measure of how close the proposed step will bring us to solving the problem. We will discuss further below, but the model count of a logical problem is the number of possible worlds (i.e. number of combinations of truth-assignments to the variables declared in the problem), for which the problem is solved. During a good proof, the variables in the problem will naturally become more constrained, leading to a reduction in model count.

## 4.1. Sampling $L_i$

We can write $L \in \mathcal{P}(\mathcal{D})$ as $L = (L_1, L_2, ..., L_{|\mathcal{D}|})$, where $\forall i \in \{1, \cdots, |\mathcal{D}|\}, L_i \in \mathcal{E}$. Since we reason one thought at a time, we consider a setting where it is only possible to sample individual elements (epistemic statements) $L_i$. Because $p(L)$ defines a joint distribution $p(L_1, L_2, ..., L_{|\mathcal{D}|})$, so that $p(L_i = l_i) = \sum_{l' \in \Omega_{S,c}} \mathbf{1}(l'_i = l_i)p(L = l')$, and $p(L_i|L_j) = \sum_{l' \in \Omega_{S \wedge l_j,c}} \mathbf{1}(l'_i = l_i)p(L = l')$, we can say by the chain rule that:

$$p(L = l) = p(L_1 = l_1)p(L_2|L_1 = l_1)$$
$$\cdots p(L_{|\mathcal{D}|} = l_{|\mathcal{D}|}|L_{1:|\mathcal{D}|-1} = l_{1:|\mathcal{D}|}) \quad (2)$$

This is also true for any ordering of $L$. Now, we describe the process of sampling by the chain rule in Equation 2.

We introduce an ordering function which maps from time-step to index of $L$: $\phi: t \mapsto \phi(t) \in \{1, 2, \ldots, |\mathcal{D}|\} \setminus \phi(1:t-1)$. We denote by $\phi(1:n)$ the choices made at each time step between $t = 1$ and $t = n$. Let us commence with a state vector $\hat{l}_0$ where every element is "Unknown". We update the state at $t = 1$ by setting $\hat{l}_1 := \hat{l}_0$, but then replacing the $\phi(1)$-th element by a random draw $l_{\phi(1)} \sim p(L_{\phi(1)})$. At the $j$-th iteration of this process, for $j > 1$, we set $\hat{l}_j := \hat{l}_{j-1}$ but then replace the $\phi(j)$-th element by a draw from $p(L_{\phi(j)}|L_{\phi(1:j-1)})$. Then, $\hat{l}_{|\mathcal{D}|} \sim p(L)$ (Eq. 2) for any selected ordering $\phi$.

## 4.2. Early Stopping

As mentioned in the problem statement, knowledge of a reasoner's beliefs over all of $\mathcal{D}$ is almost never necessary to solve a given problem $(S, c)$. In fact, if we adopt the chain-rule-sampling procedure, we can stop as soon as $T(c, S \wedge \hat{l}_t) \neq Unknown$. This is because any of the underlying values of $l$ which might be sampled by completing the process will, by the assumption of logical consistency of each $l \in \Omega_{s,c}$, necessarily entail the same truth on $c$ as this early-stopped version, which we denote $\hat{l}(l, \phi)$. This notation conveys that the vector which is a result of early stopping is a deterministic mapping from some (multiple) $l$ values, given a specific ordering $\phi$. For clarity of notation, we also describe the process of sampling starting at some intermediate state $\hat{l}_i$, going until the early-stopping condition is met as $\hat{l}(l, \hat{l}_i, \phi)$. We denote the stopping time as $\|\hat{l}(l, \phi)\|$ and $\|\hat{l}(l, l_i, \phi)\|$ respectively. Given that any ordering will preserve the underlying distribution on entailment, as discussed above (and below; see Eqs. 4 - 8), there is clearly at least one ordering $\phi_l^*$ for every $l$ which will yield the shortest stopping time $\|\hat{l}(l, \phi_l^*)\|$.

## 4.3. Commonsense Abductive Probabilistic Truth

In order to know if $c$ is True, we wish to know $P(T(c, S \wedge l) = True|S)$. The problem of determining whether a query $c$ is true, given only some context $S$, can then be written as:

$$\arg\max_{a \in \{True, False\}} p(T(c, S) = a|S), \text{ where}$$

$$p(T(c, S) = True)|S) = \sum_{l \in \Omega_{S,c}} \mathbf{1}(l \wedge S \vdash c)p(L = l)$$

$$= \sum_{l \in \Omega_{S,c}} \mathbf{1}(\hat{l}(l, \phi) \wedge S \vdash c)p(L = l) \quad (3)$$

## 4.4. Sampling $\hat{L}$ is equivalent to sampling $L$ in outcome

First, we describe the space of $\hat{l}(l, \phi)$: $\hat{l}(l, \phi) \in \hat{\Omega}_{S,c} = \{\hat{l}(l, \phi) : l \in \Omega_{S,c}\}$. Since given a full sample $l$ and an ordering $\phi$ there is clearly a single, deterministic mapping $\hat{l}(l, \phi)$, we define the random variable $\hat{L}_\phi = \hat{l}(L, \phi)$. The probability of sampling $p(\hat{L}_\phi = \hat{l}|\phi)$ is the sum of the probabilities of all $l$ for which $\hat{l}$ is the resultant early-stopped version given some ordering $\phi$: $p(\hat{L}_\phi = \hat{l}|\phi) = \sum_{l:\hat{l}(l,\phi)=\hat{l}} p(L = l)$.

From Equation 3, we have $\sum_{l \in \Omega_{S,c}} \mathbf{1}(l \wedge S \vdash c)p(L = l) = \sum_{l \in \Omega_{S,c}} \mathbf{1}(\hat{l}(l, \phi) \wedge S \vdash c)p(L = l)$. This is clear, since $\hat{l}(\cdot, \phi)$ is a function of $l$. Our goal, however, is not to sample $l$ but $\hat{l}(l, \phi)$. We must demonstrate that for any ordering $\phi$, sampling directly from $\hat{l} \sim p(\hat{L}_\phi|\phi)$, as the early-stopped chain-rule sampling process, yields the same

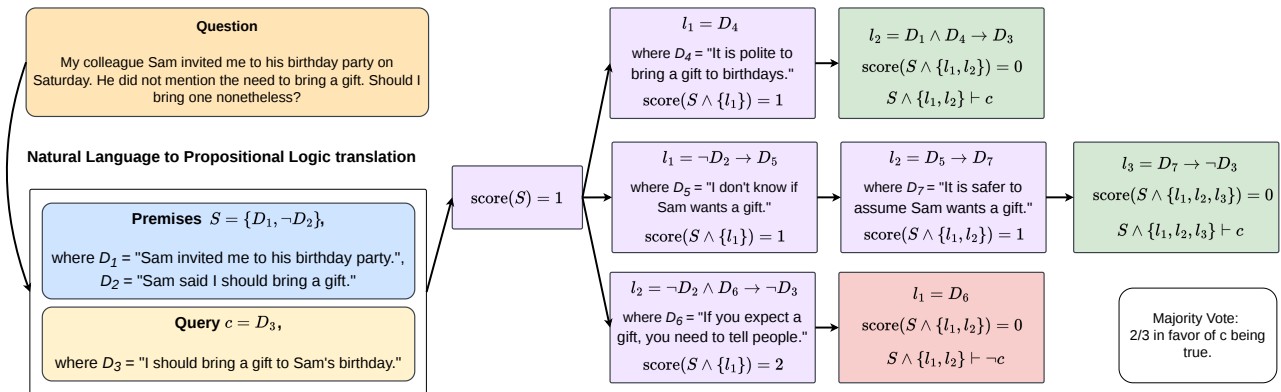

*Figure 1.* Diagram illustrating our proposed PACS algorithm. The LLM receives a question from a user which requires abductive reasoning. The LLM translates this question into premises $S$ and a query proposition $c$ whose truth value is to be determined. Ascertaining that it cannot be solved directly, the LLM then attempts to add new commonsense clauses $l_1, l_2, l_3, \ldots$, each time calling the formal logic solver to verify whether it has pinned down a value for $c$, and if not, obtain the score. We stop after a time limit and take a majority vote among the conclusions as our answer. The algorithm used to search the tree of possibilities is described in Section 5.1.

distribution over $T(c, S \wedge \hat{l})$ as sampling from $l$. Dropping the functional notation where appropriate for brevity, we have from Eq. 3:

$$\sum_{l \in \Omega_{S,c}} \mathbf{1}(l \wedge S \vdash c)p(L = l) \tag{4}$$

$$= \sum_{l \in \Omega_{S,c}} \mathbf{1}(\hat{l}(l, \phi) \wedge S \vdash c)p(L = l) \tag{5}$$

$$= \sum_{\hat{l} \in \hat{\Omega}_{S,c}} \sum_{l : \hat{l}(l,\phi) = \hat{l}} \mathbf{1}(\hat{l} \wedge S \vdash c)p(L = l) \tag{6}$$

$$= \sum_{\hat{l} \in \hat{\Omega}_{S,c}} \left[ \mathbf{1}(\hat{l} \wedge S \vdash c) \sum_{l : \hat{l}(l,\phi) = \hat{l}} p(L = l) \right] \tag{7}$$

$$= \sum_{\hat{l} \in \hat{\Omega}_{S,c}} \mathbf{1}(\hat{l} \wedge S \vdash c)p(\hat{L} = \hat{l}|\phi) \tag{8}$$

### 4.5. Finding the optimal ordering as a Markov Decision Process

The problem of selecting the ordering which minimizes stopping-time can be described as a Markov Decision Process (MDP) in which we pick the ordering at each step according to some policy $\pi(\phi(t)|\hat{l}_{t-1}, \phi(1:t-1))$:

**State at time** $t-1$: $\hat{l}_{t-1}$
**Action at time** $t$: select $\phi(t) \sim \pi(\phi(t)|\hat{l}_{t-1}, \phi(1:t-1))$, then set $\hat{l}_t := \hat{l}_{t-1}$, replacing the $\phi(t)$-th element in $\hat{l}_t$ with a draw from $p(L_{\phi(t)}|L_{\phi(1:t-1)})$.
**Reward**: $-\mathbb{E}_{L_{\phi(t)}|L_{\phi(1:t-1)}}\mathbb{E}_{L|L_{\phi(1:t)}}[\|\hat{l}(L, \hat{l}_t, \phi)\|]$.

We need to find a policy for selecting $\phi(t)$, given the state $\hat{l}_{t-1}$, so that the reward is maximized (the expected number of future steps is minimized). Until now, we have assumed the ordering $\phi$ is non-adaptive (i.e. $\phi(t)$ is dependent only on $\phi(1:t-1)$). The previous arguments also hold for an adaptive $\phi(t)$ (i.e., $\phi(t)$ is also dependent on $\hat{l}_{t-1}$), so long as the chosen adaptive policy is deterministic, so that it still leads to a deterministic mapping from $L$ to $\hat{L}$.

### 4.6. Identifying a surrogate

In expectation, the optimal $\phi(t)$, given $\hat{l}_{t-1}$, satisfies:

$$\phi_{\hat{l}_{t-1}}^*(t) = \arg\min_{\phi(t)} \mathbb{E}_{L_{\phi}(t)|L_{\phi(1:t-1)}} \Bigg[ \tag{9}$$
$$\mathbb{E}_{L|L_{\phi(1:t)}} \left[ \|\hat{l}(L, \hat{l}_t, \phi_{\hat{l}_{t-1}}^*(t+1)\| \right] \Bigg]$$

For brevity, we write $V(\hat{l}_t) = \mathbb{E}_{L|L_{\phi(1:t)}} \left[ \|\hat{l}(L, \hat{l}_t, \phi_{\hat{l}_{t-1}}^*(t+1)\| \right]$. So, we select the $\phi(t)$ that minimizes

$$\mathbb{E}_{L_{\phi(t)}|L_{\phi(1:t-1)}} \left[ V(L_{\phi(t)}) \right] = \sum_{l_{\phi(t)} \in \mathcal{E}} V(\hat{l}_t)p(l_{\phi(t)}|l_{\phi(1:t-1)})$$

Evaluating $V(\hat{l}_t)$, however, requires recursively evaluating this function over all future steps until all paths terminate according to the stopping criterion. To find a tractable alternative $\widetilde{V}$, we can observe that any path will, **in worst case**, eventually reach a state in which each variable of the problem is fully constrained. Once the problem is fully constrained, the next step that further constrains the variables in the problem is guaranteed to be a stopping-point. Given $\hat{l}_t$, we name the set of all of these fully-constraining states, which can be reached from $\hat{l}_t$, as $\sigma$. There are $\#\sigma$ different states in the set $\sigma$. To reach each $\sigma$, we require that sufficient propositional beliefs are added to $\hat{l}_t$, so that each of the variables in the problem is constrained. In the worst case, each step constrains only one new variable. Therefore, any $\sigma_i \in \sigma$ is reachable from state $\hat{l}_t$ in a number of steps equal to the difference between the num-

ber of variables in the problem and the number of already-constrained-at-$\hat{l}_t$ variables in the problem. We write this difference $v_{S \wedge \hat{l}_t} - b_{S \wedge \hat{l}_t}$, where $v$ and $b$ are the number of variables in the question, and the number of variables that are already fully constrained by $S \wedge \hat{l}_t$, respectively. So,

$$\mathbb{E}_{L|L_{\phi(1:t)}}[\|\hat{l}(L, \hat{l}_t, \phi^*_{\hat{l}_{t-1}}(t+1))\|] \leq$$

$$\sum_{\sigma_i \in \sigma} \left[ t + v - b + V(\sigma_i) p(\sigma_i | l_{\phi(1:t)}) \right] = \quad (10)$$

$$\sum_{\sigma_i \in \sigma} \left[ t + v - b + V(\sigma_i) \right] \quad (11)$$

Now, since each $\sigma_i$ defines a complete specification over the problem, $\forall \sigma_i \in \sigma, [V(\sigma_i) \approx 1]$. If we eliminate the terms which are constant for all choices $\phi(t)$ at step $t$, we have that by identifying $\arg\min_{\phi(t)} \left[ \#\sigma \times (v-q) \right]$ we maximize a lower bound of our defined reward. We therefore specify the score that we propose to minimize greedily as

$$Score(\phi(t)|S) = \#\sigma^{S \wedge l_{\phi(1:t)}} \times (v_{S \wedge \hat{l}_t} - b_{S \wedge \hat{l}_t}) + 1. \quad (12)$$

The most noteworthy element of this derivation is that (under a strong assumption and some approximations) we have succeeded in bounding an expectation that would normally require knowledge of the full distribution $p(L)$, using only knowledge of the current state $\hat{l}_t$.

# 5. Evaluation

**Experimental Setting**   We are given an abductive propositional logic problem in textual form and provided with a large language model and a SAT solver. The task is to determine whether the target query is true or false given the premises and some additional commonsense propositions which must be found. Four annotated examples are provided, intended for few-shot prompting of the LLM. In particular, the task is inference-only and no training phase is involved. We evaluate performance based on the number of correctly answered questions on a test dataset.

**Datasets: Abductive-FOLIO.** We build upon FOLIO, a dataset of true–false questions designed to test logical and commonsense reasoning, adapting it to an abductive reasoning setting. This adaptation is performed by human annotators, who modify the text by replacing certain phrases so that the corresponding predicate symbols in the first-order logical interpretation change, while preserving the original semantic meaning. For instance, the term "Professor" may be replaced by "Teacher" in parts of a problem, introducing a missing but necessary logical relation

$$\forall x, \left[ professor(x) \leftrightarrow teacher(x) \right].$$

**CosmosQA.** CosmosQA is originally a multiple-choice reading comprehension dataset. We convert it into a true–false format by randomly selecting either the correct answer or one of the incorrect answers and assigning a label of *True* or *False* accordingly.

**QUAIL.** QUAIL is similarly a multiple-choice reading comprehension dataset. We process it using the same conversion procedure as CosmosQA.

We select FOLIO due to its strong grounding in formal logical reasoning, and CosmosQA and QUAIL because they naturally involve abductive inference, illustrating the broad applicability of logical reasoning across diverse domains.

**Baselines** We include two neural baselines: Chain of Thought (COT) (Wei et al., 2022) and Self-Consistency (SC) (Wang et al., 2023); three neurosymbolic baselines: Logic of Thoughts (LoT) (Liu et al., 2024), LLM-Tres (Toroghi et al., 2024) and ARGOS (Cotnareanu et al., 2026); and a heuristic search: Beam-IF (Wang et al., 2025).

**A brief discussion of the baselines under our framework** Under our proposed framework, COT can be seen as taking a one-sample Monte-Carlo approximation of $p_{S,c}(\hat{L})$, via a simple LLM generative call. SC can be seen in the same way, but with more than one sample in the Monte-Carlo approximation (in this work we use 20 for self-consistency). While SC improves robustness through sampling compared to COT, it still relies entirely on the LLM's implicit reasoning process to solve the minimization problem. This is unrealistic, however, and often leads to overthinking and overconfidence issues (Cuesta-Ramirez et al., 2025).

LoT operates by augmenting the input logic $S$ with alternative linguistic realizations of statements already entailed by $S$ (e.g., "$A \Rightarrow B$" augmented with its contrapositive "$\neg B \Rightarrow \neg A$"). Given this augmented context, the method then applies self-consistency by sampling multiple reasoning traces and aggregating their predictions. From our perspective, the augmented set $\{l_1, \ldots, l_m\}$ can be viewed as instantiating different *redundant realizations* of the latent logical structure underlying $S$.

LLM-Tres iteratively generates and adds new propositions to the problem. At time $t$, the generated proposition $l_t$ is guaranteed to entail one of the already-instantiated propositional variables of the problem. This design ensures that the length of any generated proof is bounded by the number of variables in the problem's logic. Next, the unconditional probability $p(l_t)$ is evaluated, so that once the query is entailed, the full probability of the generated proof is computed as $p(l_{1:v}) = \prod_{t=1}^{v} p(l_t)$. This approach is problematic, however, because it implicitly assumes that each $l_t$ is probabilistically independent. Finally, multiple proofs are generated via search, and the most probable proof according to $p(l_{1:v})$ is used to produce a final decision on $c$.

ARGOS is designed to rely strongly upon the assump-

tion that commonsense is monolithic and logically consistent, meaning that the addition of any information deemed commonsensical will not contradict the problem. So, the approach in ARGOS is to repeatedly sample the LLM for new commonsense clauses, with little consideration about whether clauses are useful. This leads to more-than-necessary computation, and occasional confusion of the LLM as these clauses are added to the prompt; useless-but-not-wrong information will sometimes mislead the LLM during COT generation.

IF-Beam designs a score on sentences in a proof, prioritizing steps which are both novel and informative. Intuitively, we can see how this might be a reasonable approach to getting close to the minimization: sentences which do not add to the entailment of the query (uninformative), and sentences which dwell on the same concepts as previous steps (not novel) are clearly unlikely to contribute to the decrease in expected solve-time.

**Cost** For independently sampled methods (LoT, Self-Consistency), we ensure an equal number of samples. For search-based methods (ours, If-Beam), we ensure the same number of samples per step and the same beam-width. LLM-Tres and COT need few LLM calls relative to the other methods and their costs are left unconstrained.

### 5.1. Implementing Our Proposal

We use an LLM as the conditional sampler of $l_i$. For the implementation of our method, we require the logical translation of the textual thought. We implement this by having the LLM generate first its thought as text and then its translation, via few-shot prompting. We compute a Monte Carlo estimate of Eq. 3 by collecting samples of $\hat{l}$. In order to collect multiple samples of $\hat{l}$, we sample $n$ possible thoughts at each step and keep not just the highest-scoring sample at each step but the $m$ highest scoring steps. At the next step we sample for each of these. This is computationally much cheaper than $m$ separate greedy search procedures, since each of the $m$ samples shares most of its computation with the other samples. Generation of a single thought is much shorter than the length of the full reasoning input + partial COT. Each time we sample a "Final Answer" from the LLM, we save the full path. When time $t$ exceeds the limit we set, we end the search. Once the full search terminates, we evaluate Equation 3 via Monte Carlo approximation over the set of all saved paths, treating each path as a sample of $L$. Effectively, this cost-saving implementation of our search is very similar to a beam-search. There are, however, two important differences. The first is that we retain **all** completed searches and not just the highest-scoring ones. The second is that the final decision is not made according to the single highest-scoring path, but according to the majority-vote of all extracted paths.

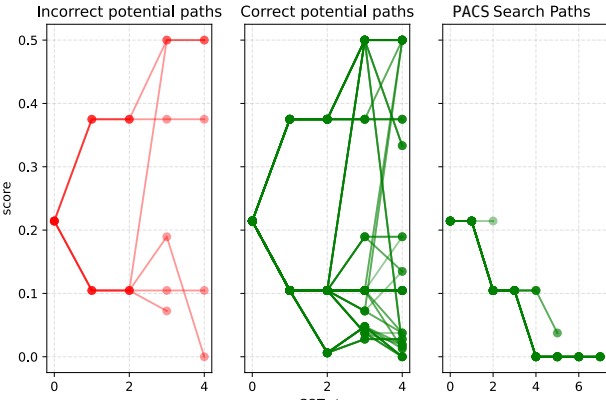

*Figure 2.* The (normalized) score progression of LLM sampled and PACS sampled paths. On the left and middle, we generate paths exhaustively taking 3 sample next-thoughts at each node. On the left, we show the model-count-based scores for incorrect paths and in the middle for the correct ones. We find no discernible difference between correct and incorrect score behaviour, indicating unfaithful LLM reasoning. On the right however, PACS shows that when using our minimization approach we find well-minimized paths which are **all correct**. This shows that the LLM itself can not properly solve the minimization we describe, and eventually gives up with a guess. This is validated by analysis of the text generated over the paths, shown in Figure 3.

### 5.2. Main Accuracy Results

In Table 1, we find that PACS performs exceptionally well on FOLIO and QUAIL, improving on the heuristic beam-search approach by 8% and 6% respectively. On CosmosQA, PACS performs less impressively, showing similar performance to the text-based heuristic search. We attribute this weaker performance to a combination of CosmosQA having typically simpler questions which do not require the same degree of multi-hop reasoning as the other datasets (effective answer extraction is more important). We also attribute this result to the fact that in our implementation, we depend on either Llama 3 8B or Llama 3.3 70B to translate each of its COT steps to logic as it progresses. In practice, mistakes are frequently made. These of course affect our ability to apply the score minimization effectively. Via manual inspection, we found that the LLM made translation mistakes more frequently on CosmosQA than on other datasets. Ideally, a logical translation model could be fine-tuned to greatly improve the translation and address this issue; Yang et al. (2024) demonstrate that fine-tuning even Llama 2 8B on logical translation tasks leads to GPT4-level performance. We find similar relative performances on Llama 3.3 70B, demonstrating that our findings scale to larger and more recent language models.

Table 2 shows on average, how frequently a method's set of sampled reasoning traces includes **not a single** correct final answers. Of course, being binary in output, this statistic is very small for each method which samples multiple traces. With that said, we still find it informative as a measure of

*Table 1.* Accuracy using Llama 3-Instruct 8B and Llama 3.3 70B. PACS strongly outperforms all existing methods despite our naive assumptions, demonstrating both the utility and validity of the underlying framework. Best-performing accuracies are bolded. 95/5% confidence intervals are indicated in small font next to each accuracy value.

| | FOLIO | CosmosQA | QUAIL | Disamb-QA | FOLIO | CosmosQA | QUAIL | Disamb-QA |
|---|---|---|---|---|---|---|---|---|
| PACS (ours) | $\mathbf{82\%}^{85}_{79}$ | $82\%^{85}_{78}$ | $\mathbf{80\%}^{84}_{76}$ | $\mathbf{88\%}^{93}_{83}$ | $\mathbf{88\%}^{91}_{85}$ | $\mathbf{91\%}^{93}_{88}$ | $\mathbf{84\%}^{87}_{81}$ | $\mathbf{93\%}^{96}_{89}$ |
| LoT | $69\%^{73}_{65}$ | $76\%^{80}_{72}$ | $56\%^{60}_{51}$ | $71\%^{78}_{64}$ | $70\%^{74}_{66}$ | $85\%^{88}_{82}$ | $72\%^{76}_{68}$ | $78\%^{84}_{72}$ |
| LLM-Tres | $63\%^{67}_{59}$ | $51\%^{55}_{47}$ | $56\%^{60}_{51}$ | $56\%^{63}_{48}$ | $63\%^{67}_{59}$ | $51\%^{55}_{47}$ | $56\%^{60}_{51}$ | $56\%^{63}_{48}$ |
| COT | $68\%^{72}_{64}$ | $75\%^{79}_{71}$ | $66\%^{70}_{62}$ | $74\%^{80}_{67}$ | $73\%^{77}_{69}$ | $88\%^{91}_{85}$ | $72\%^{76}_{68}$ | $89\%^{93}_{84}$ |
| SC-20 | $71\%^{75}_{67}$ | $80\%^{83}_{76}$ | $70\%^{74}_{66}$ | $81\%^{86}_{74}$ | $77\%^{81}_{73}$ | $90\%^{93}_{87}$ | $75\%^{79}_{71}$ | $92\%^{95}_{88}$ |
| ARGOS | $78\%^{81}_{75}$ | $82\%^{85}_{78}$ | $73\%^{77}_{69}$ | $80\%^{86}_{74}$ | $80\%^{83}_{76}$ | $90\%^{93}_{87}$ | $80\%^{83}_{76}$ | $83\%^{88}_{77}$ |
| If-Beam | $74\%^{78}_{70}$ | $82\%^{85}_{78}$ | $74\%^{78}_{70}$ | $72\%^{78}_{65}$ | $78\%^{82}_{74}$ | $88\%^{92}_{86}$ | $82\%^{85}_{78}$ | $88\%^{92}_{83}$ |

*Table 2.* % of problems for which the correct answer was never sampled, using Llama 3-Instruct 8B. PACS demonstrates far greater exploratory capabilities than the existing methods, finding at least one correct reasoning path more frequently than the baseline methods. Considering that PACS allows for the heterogeneity of commonsense, it is not surprising that it would see greater diversity in outcome.

| | FOLIO | CosmosQA | QUAIL |
|---|---|---|---|
| PACS (ours) | **3%** | **1%** | **2%** |
| LoT | 8% | 5% | 5% |
| LLM-Tres | N/A | N/A | N/A |
| COT | N/A | N/A | N/A |
| SC-20 | 6% | 3% | 4% |
| ARGOS | N/A | N/A | N/A |
| If-Beam | 16% | 14% | 21% |

diversity of output and of each method's ability to address **gross LLM over-confidence**. We find that PACS greatly reduces the number of problems which see zero traces with correct final answers, halving the statistic with respect to the next-best method on each dataset. Given that PACS is specifically designed to assume heterogeneity (diversity) in the reasoner population, this result is not surprising. Most notably, we find that despite the beam-search baseline's strong performance on accuracy, it suffers from exceptionally low diversity in output.

### 5.3. Example: Comparing All Tree Paths To Those Selected by PACS

Figure 2 shows, on a specific example from FOLIO, how the application of our proposed search procedure to LLM reasoning eliminates LLM error by properly searching for the minimal explanation. In the left and middle panels, we see that when the LLM is allowed to generate a full answer path without guidance, its scores are often increasing. Most

*Table 3.* Average Lengths of Paths Found By Search, Llama 3 8B

| | FOLIO | CosmosQA | QUAIL |
|---|---|---|---|
| If-Beam | 6.7 | 5.4 | 5.2 |
| PACS | **4.6** | **3.7** | **4.1** |

notably, there is little observable difference in the score trend of correct and incorrect answers, demonstrating unfaithful behavior. In Figure 3, we show two example paths generated by the LLM for this problem, highlighting that for nearly identical paths the LLM chooses a different final answer. This result demonstrates an important reason, aside from cost, to reduce the number of steps. As the chain of thoughts increases in length, the LLM's ability to model real reasoning clearly diminishes; its training likely biases it towards concise answers. So, as the chain draws longer, the LLM becomes more likely to "guess" a final answer. If the already-chosen steps in the chain are of poor quality, this guess is likely to be unreliable.

### 5.4. Average Path Length

For the two search-based methods (ours and If-Beam), we compare the average step-length. Table 3 shows that If-Beam has a higher average path-length than PACS, indicating a fundamental failure to "plan ahead" and find the minimal solution, resulting in lower overall accuracy on FOLIO and QUAIL (which require more in-depth reasoning).

### 5.5. Comparison With Thinking Models

Here, we compare the proposed method with more modern "Thinking" models, which are trained to generate a preamble to the COT, in which the model generates a plan, or sketch, of the output. We observe that these models tend to outline the full reasoning process in an un-structured

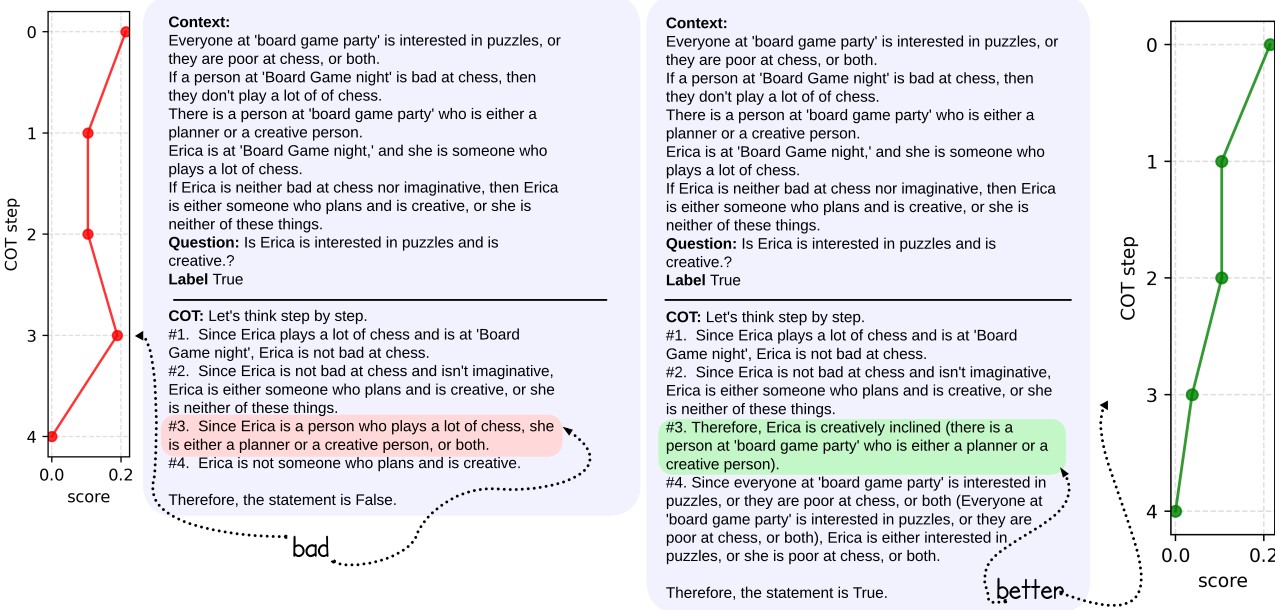

*Figure 3.* Comparison of two very similar reasoning paths with opposite answers. On the left, we see a reasoning path in which, at step 3, a step which is not necessarily false but increasing in score is introduced. This clearly "throws off" the LLM, as its next step is simply the final (wrong) answer. On the right, however, we see that step 3 pushes the score down, bringing the path closer to a logically valid final answer. The LLM continues reasoning faithfully, answering correctly in the end.

*Table 4.* Self-Consistency with 120B thinking model, compared to PACS with 70B and 8B non-thinking models.

|  | FOLIO | QUAIL |
|---|---|---|
| SC OSS-GPT-120B | 82% | 80% |
| PACS Llama 3.3 70B | **88%** | **84%** |
| PACS Llama 3 8B | 82% | 80% |

way here. This observation makes the proposed method (and any step-level search procedure) difficult to implement in Thinking models. However, since such models are widely considered to be the foremost reasoning models, there might be concern that such procedures will become obsolete. In Table 4, we compare our method with GPT-OSS-120B (an open-source 120B Thinking model released in August 2025). We find that even using an older, smaller, non-thinking model, our proposed method outperforms modern Thinking models substantially. Most surprisingly, PACS Llama 3 8B shows similar performance to SC OSS. This demonstrates perhaps that reasoning is more efficiently achieved by designing concrete, principled search algorithms rather than fine-tuning the LLM directly.

## 6. Conclusion

We propose in this work a system by which we might understand how a society of humans reasons in commonsensical, abductive settings. The summary of the proposed system is that to answer such problems, we must not ask ourselves *"which is correct?"*, but *"which would most say is correct?"*; hence, the sum over sample rationales. Also, in order to solve a problem, we do not ask ourselves *"what are some true statements?"*, but *"what are some true statements which will help me solve this problem?"*; hence, the minimization. While both of these statements seem self-evident, the literature has yet to adopt representative perspectives. We demonstrate empirically the value of the proposed framework, and the truth of its underlying assumptions: despite implementing it fairly naively (e.g., a uniform model of solutions) we measured impressive performance on abductive logical reasoning problems.

## 7. Limitations

The primary limitation of this work is the need to model $p(\sigma_i | l_{\phi(1:t)})$ as uniform. This is clearly an unrealistic model. While it is likely that a more involved model of this distribution may be possible with an LLM, this was deemed outside the scope of this paper. Additionally, this work relies on a pre-trained LLM for translation of natural language to logic, introducing logical errors as a secondary factor of error during empirical evaluation. Some work has shown that a small, relatively weak LLM can be fine-tuned to perform logical translation extremely well. Adapting such a tool for our purpose, however, was also deemed out of scope for this work.

## Impact Statement

This paper presents work whose goal is to advance the field of Machine Learning. There are many potential societal consequences of our work, none which we feel must be specifically highlighted here.

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

# A. Prompts

Our implementation of PACS requires thought-by-thought logical translation; each textual thought expressing reasoning must be converted to logic so that we can compute our ranking score.

*Table 5.* Example PACS COT

```
Prompt:
Here is some context:
# The girl is the narrator's daughter.
my_daughter(her)
# The narrator does not walk her to the bus or hold her hand.
walk_out_alone(her)
# If the daughter walks out alone, she is independent.
Implies(walk_out_alone(her), independent(her))
# Question: Why might I not walk her to the bus stop ?
# Students answer: Because she is getting old enough to walk to the bus stop
with her friends without my supervision.
# Instruction: determine if the student's answer is True or False.
return Implies(old_enough(her), walk_out_alone(her))
Let's think step by step:
#1.
```
**LLM generation:**
```
Since the daughter walks out alone, she is becoming independent.
Implies(walk_out_alone(her), independent(her))
#2. Children become independent as they get older.
Implies(independent(her), getting_older(her)) #3. When a child gets older, they
eventualy become old enough
Implies(getting_older(her), old_enough(her))
#4. Therefore, the student's answer is correct.
return True
```

