# OpenReview forum: "Abductive Reasoning with Probabilistic Commonsense"
_ICML.cc/2026/Conference — ICML 2026 regular_

### Official Review · Reviewer_ipcA · 2026-02-14

**Soundness:** 3
**Presentation:** 3
**Significance:** 3
**Originality:** 3
**Overall Recommendation:** 4
**Confidence:** 3

**Summary:**

This paper proposes a probabilistic framing of abductive commonsense reasoning that treats missing commonsense as variable across people, not a single universally agreed set of facts. The method, PACS, uses an LLM plus a SAT solver to sample plausible commonsense additions, stop early once the query or its negation is entailed, and aggregate results via a Monte Carlo vote. It also uses a search procedure with a derived score to prioritize steps that quickly constrain the problem. Experiments on Abductive-FOLIO, CosmosQA, and QUAIL show higher accuracy than CoT, self-consistency, and several neurosymbolic and search baselines.

**Compliance With Llm Reviewing Policy:**

Affirmed.

**Final Justification:**

The paper presents an interesting probabilistic view of abductive commonsense reasoning, with a coherent solver-backed framework and solid empirical gains. The rebuttal addressed my main concern by clarifying that the score derivation does not require the uniformity assumption, and the added translation analysis was helpful. Some dataset-alignment concerns remain, but I increase my score to weak accept after the discussion.

**Key Questions For Authors:**

1. How sensitive is PACS to the uniformity assumption used in the surrogate score, and what happens if you replace the score with simpler heuristics or learned scoring?
2. Can you provide an ablation where the logic translation is made near-perfect, or at least measured, to quantify how much performance is capped by translation errors?
3. For CosmosQA and QUAIL, how well does the *abductive truth as population vote* framing match the dataset labels, and do you have any human study or calibration evidence?

**Limitations:**

Yes, the limitations are well discussed in Section 7.

**Strengths And Weaknesses:**

Overall, I like the direction and the paper is fairly coherent, but I believe that the core modeling assumptions and eval alignment need more tightening.

Strengths:
- I agree that the probabilistic perspective on abductive commonsense is a clean way to make “probability of truth” more interpretable than ad-hoc confidence.
- The early stopping and solver-backed entailment checks give a concrete mechanism to reduce overlong, error-prone CoT traces.
- The method is evaluated against a reasonably broad set of baselines, including neurosymbolic and search-style approaches. The authors also use metrics beyond accuracy, like zero-correct-trace rate and average path length, which helps diagnose why the approach can help.

Weaknesses:
- The score derivation relies on strong approximations, especially a near-uniform assumption over reachable fully constraining states, and it is unclear how sensitive results are to this.
- The pipeline depends on natural language to logic translation quality, but there is no controlled ablation isolating translation errors from the claimed search and aggregation benefits. I guess maybe modern LLMs can handle this really well? But I'm not empirically confident about this.
- Dataset alignment with the “majority of humans would judge true” objective is not validated, especially for converted CosmosQA and QUAIL where labels come from a construction procedure.

---

> ### Author Rebuttal · Authors · 2026-03-31
>
> We are grateful for the reviewer's thought-provoking and crucial
> feedback and questions. We are also grateful for the author's
> acknowledgment of our approach's value in interpreting truthfulness as a
> probability, of the concrete utility of our proposed approach, and the
> robustness of our experimental setting.
>
> ## Weaknesses
>
> 1.  *The score derivation relies on strong approximations, especially a
>     near-uniform assumption over reachable fully constraining states.*\
>     Your comment made us think harder about when the assumption might
>     not hold, and made us realize that it is, in fact, not even
>     required. Indeed, we already observe in the text that
>     $V(\sigma_i)= 1$ for $l \in \Omega_{S,c,l_{\phi(1:t)}}$ such that
>     $S \land \sigma_i \land l_{\phi(1:t)}$ fully constrains the formula.
>     This implies that
>     $$\sum_{\sigma_i \in \sigma}[t + v - b + V(\sigma_i)p(\sigma_i \mid l_{\phi(1:t)})]$$
> $$=\sum_{\sigma_i \in \sigma}[t + v - b] + \sum_{\sigma_i \in \sigma}p(\sigma_i \mid l_{\phi(1:t)})]$$
> =# $\sigma^{S \land l_{\phi(1:t)}} (v_{S \land \hat l_t} - b_{S \land \hat l_t}) + 1$,
>     which is exactly our score function, up to an irrelevant constant.
>     Thus the uniformity assumption was not even needed and we will
>     remove it from the text.
>
> 2.  *The pipeline depends on natural language to logic translation
>     quality, but there is no controlled ablation isolating translation
>     errors.*\
>     We filtered out failed translations and evaluated only correctly
>     translated instances. On FOLIO with Llama 3 8B, accuracy increased
>     from 82% to 84%, indicating translation errors account for a small
>     portion of total error, consistent with prior work, e.g. \[**1,
>     2\]**.\
>     **\[1\]** Yang et al. (2024). Harnessing the power of large language
>     models for natural language to first-order logic translation.
>     Proceedings of ACL 2024.\
>     **\[2\]** Cotnareanu et al. (2026). A Balanced Neuro-Symbolic
>     Approach for Commonsense Abductive Logic. ArXiv:2601.18595.
>
> 3.  *Dataset alignment with the "majority of humans would judge true"
>     objective is not validated, especially for converted CosmosQA and
>     QUAIL where labels come from a construction procedure.*\
>     As discussed above, we are unable to conduct a human study. However,
>     we find that PACS confidence correlates with question ambiguity (and
>     we expect ambiguity to correlate with true human voter confidence).
>     Examples:\
>     Votes: 24 True, 137 False. Confidence = 85%:\
>     Context: The educator was meeting with a student to discuss his
>     grading policy.\
>     Query: It was the student's grading policy\
>     Label: False.\
>     Votes: 76 True, 74 False. Confidence: 51%:\
>     Context: The customer asked the salesperson if she could send the
>     prices of each product.\
>     Query: The salesperson would send the prices\
>     Label: True.\
>     The first is logically clear, while the second requires insight into
>     what may happen in the future and is therefore more ambiguous.
>
> ## Questions
>
> 1.  *How sensitive is PACS to the uniformity assumption?*\
>     As described above, we will remove this assumption from our
>     analysis.
>
> 2.  *What happens if you replace the score with simpler heuristics or
>     learned scoring?*\
>     The If-Beam baseline uses the same beam search but a linguistic
>     heuristic score, resulting in lower accuracy and longer reasoning
>     paths.
>
> 3.  *an you provide an ablation where the logic translation is made
>     near-perfect, or at least measured, to quantify how much performance
>     is capped by translation errors?*\
>     Filtering out failed translations increases accuracy from 82% to 84%
>     on FOLIO (Llama 3 8B), suggesting translation errors are not the
>     main bottleneck.
>
> 4.  *How well does the abductive truth as population vote framing match
>     dataset labels?*\
>     Please see our response to weakness 4 for some examples showing that
>     PACS confidence is typically well-correlated with (subjective) true
>     ambiguity and thus with the human vote (more ambiguous problems will
>     likely result in less one-sided human votes).

---

> > ### Author Rebuttal · Reviewer_ipcA · 2026-04-02
> >
> > Thanks to the authors for the helpful rebuttal. The response resolves my main concern about the score derivation by clarifying that the uniformity assumption is not actually required, and the added translation analysis is useful. I still find the dataset-alignment evidence somewhat indirect, but overall my key concerns are sufficiently addressed. I will increase my score accordingly.

---

> > > ### Author Response · Authors · 2026-04-07
> > >
> > > We are grateful to the reviewer for the engaging and productive rebuttal discussion. We are glad that you have found our response to be useful. While the human data-alignment question is a difficult one to address, we hope that the results provided have added some credibility to our claims and experimental setting.

---

### Official Review · Reviewer_4cRi · 2026-03-12

**Soundness:** 3
**Presentation:** 3
**Significance:** 3
**Originality:** 3
**Overall Recommendation:** 4
**Confidence:** 3

**Summary:**

Main contributions: the paper introduces a novel method of combining an LLM and a formal solver to solve reasoning tasks, with the focus being on sampling different commonsense assumptions, and producing a final decision by aggregating across the different samples. Experiments on three datasets demonstrate that the method surpasses the performance of other neurosymbolic methods and also classic chain-of-thought reasoning.

**Compliance With Llm Reviewing Policy:**

Affirmed.

**Final Justification:**

This paper is an interesting contribution based on a principled approach. The rebuttal shows the authors can upgrade the paper if needed.

**Key Questions For Authors:**

The caption of Table 1 states that "statistically significant best-performing accuracies are bolded", but it is not clear which test was performed to measure statistical significance. It would be good to clarify this.

How did you choose the datasets FOLIO, CosmosQA and QUAIL? Please clarify why you chose these three datasets, since there seems to be a relatively large ecosystem of datasets designed to measure logical reasoning abilities (e.g. P-FOLIO, LogicBench, PC-FOL, RuleTaker)?

**Limitations:**

The authors have addressed limitations.

**Strengths And Weaknesses:**

The paper is well-written and clear. The related work section is extensive and up-to-date. The key idea of aggregating across different commonsense assumptions, i.e. asking "which would most say is correct" instead of "what is correct", is interesting and the paper treats this topic in a novel way. The topic in general is is significant and interesting to a wide variety of AI-researchers beyond narrow niches.

Concerning points of possible improvement, the experiments were performed with only a single, small LLM: Llama 3-Instruct 8B. This model was published in April 2024, so since one and a half years is a longish time in a field advancing as rapidly as LLMs are, one may ask about generalizability in relation to newer models. Also, including more extensive experiments would provide a stronger case for the paper.

Fairness of the COT baseline can be discussed, since the studied model is explicitly not trained to use chain-of-thought thinking, unlike other similar, open-weight models (e.g. Phi-4-reasoning, Qwen3-Next-80B-A3B-Thinking, or GPT-OSS-120b).

---

> ### Author Rebuttal · Authors · 2026-03-31
>
> We thank the reviewer for providing such valuable and considered
> feedback. We are also happy to see that the reviewer found that our
> paper was well-written, that our related-works section was complete and
> up-to-date, that our approach was interesting and novel, and that our
> work is widely impactful.
>
> ## Weaknesses
>
> 1.  *Only a single, small LLM is used*\
>     We re-ran Table 1 experiments on the larger and newer Llama 3.3 70B
>     on FOLIO and QUAIL. PACS performs strongly: on FOLIO, PACS achieves
>     88% accuracy vs. Self-Consistency 77%, LoT 70%, ARGOS 80%, and
>     IF-Beam 78%. On QUAIL, PACS achieves 84%, vs. SC 75%, LoT 72%, ARGOS
>     80%. We also tested Self-Consistency on GPT-OSS-120B, a COT-trained
>     model, which achieved 82% on FOLIO and 80% on QUAIL. PACS on Llama
>     3.3 70B outperforms SC on the larger thinking model. We also note
>     that the large and modern thinking model performs similarly to PACS
>     on Llama 3 8B.
>
> 2.  *The LLM used is old; one may ask about generalizability in relation
>     to newer models*\
>     Our access to open-source LLMs is limited by API availability and
>     GPU resources. We therefore added Llama 3.3 70B (Dec 2024). Results
>     show PACS continues to perform strongly on newer/larger models. We
>     also compared against Self-Consistency on GPT-OSS-120B; PACS on
>     Llama 3.3 70B achieves higher accuracy despite the model being
>     smaller and older. Notably, the thinking model performs similarly to
>     PACS on Llama 3 8B. Thinking models are also less suitable for
>     thought-wise methods because much reasoning occurs in hidden
>     thinking tokens rather than explicit steps. Nevertheless, we have
>     reported results in our response to W1 which demonstrate that the
>     method scales to larger, (slightly) newer LLMs, and still
>     outperforms the most modern open-source LLMs.
>
> 3.  *Including more extensive experiments would provide a stronger
>     case.*\
>     We added a new dataset, disambiguation_qa (Big-Bench Hard). Results
>     show strong performance due to the method's explicit handling of
>     ambiguity. Updated main results including disambiguation_qa are
>     provided in the new table (link). Across datasets, results on the
>     new LLM show the method generalizes well and remains the most
>     performant among tested approaches.
>
> 4.  *Fairness of the COT baseline can be discussed, since the studied
>     model is explicitly not trained to use chain-of-thought thinking,
>     unlike other sim- ilar, open-weight models (e.g. Phi-4-reasoning,
>     Qwen3-Next-80B-A3B- Thinking, or GPT-OSS-120b*\
>     We evaluated Self-Consistency on GPT-OSS-120B. On FOLIO it achieves
>     82% vs. PACS 88% on Llama 3.3 70B (and 82% on PACS with Llama 3 8B).
>     On QUAIL it achieves 80% vs. PACS 84%. Thus, the 120B COT-trained
>     model performs similarly to PACS on Llama 3 8B, while PACS on Llama
>     3.3 70B outperforms it despite having fewer parameters.
>     Additionally, we found that OSS-120B had similar performance to PACS
>     on Llama 3 8B. Thinking models also require significantly more
>     tokens. We find this result encouraging, as it supports the
>     neurosymbolic motivation.
>
> ## Questions
>
> 1.  *The caption of Table 1 states that "statistically significant
>     best-performing accuracies are bolded", but it is not clear which
>     test was performed to measure statistical significance. It would be
>     good to clarify this.*\
>     We will amend the caption to specify: "Statistically significant
>     best-performing accuracies, according to the Wilcoxon pairwise
>     signed-rank test with $p < 0.05$, are bolded."
>
> 2.  *How did you choose the datasets FOLIO, CosmosQA and QUAIL? Please
>     clarify why you chose these three datasets, since there seems to be
>     a rel atively large ecosystem of datasets designed to measure
>     logical reasoning abilities (e.g. P-FOLIO, LogicBench, PC-FOL,
>     RuleTaker)?*\
>     We chose FOLIO for its linguistic complexity, reasoning depth, and
>     widespread use in related work. To evaluate robustness to real-world
>     language and ambiguity (beyond FOLIO's synthetic ambiguity), we
>     chose CosmosQA and QUAIL, which are colloquial and challenging
>     reading comprehension datasets. We also added disambiguation_qa
>     (BigBench Hard) due to its short contexts and ambiguity-focused
>     design; results show large gains, highlighting the importance of
>     explicit ambiguity handling. We considered ProntoQA and ESNLI but
>     found them insufficiently challenging for modern methods. P-FOLIO
>     was unnecessary given FOLIO. LogicBench and RuleTaker are synthetic
>     and linguistically simple, making them less suitable for evaluating
>     realistic multi-hop reasoning. PC-FOL is promising, but adopting it
>     would require additional human annotation to create abductive
>     requirements, and it was not available at the time of submission.

---

> > ### Author Rebuttal · Reviewer_4cRi · 2026-04-03
> >
> > I thank the authors for addressing all the issues raised. Furthermore, the authors seem to have worked hard to adjust the issues, where possible. This includes, for example, expanding the scope of the experiments.

---

> > > ### Author Response · Authors · 2026-04-07
> > >
> > > We thank the reviewer for acknowledging the hard work put into addressing the valuable criticism provided.

---

### Official Review · Reviewer_6UFs · 2026-03-13

**Soundness:** 3
**Presentation:** 2
**Significance:** 3
**Originality:** 3
**Overall Recommendation:** 4
**Confidence:** 2

**Summary:**

This paper introduces a probabilistic method to inferring real-world knowledge that can be ambiguous. The method uses none deterministic commonsense statement to reach a probabilistic consensus to inform the output of the models. They compare this approach to more traditional commonsense reasoning strategies such as CoT, Self-consistency, and neuro-symbolic ones like Logic of Thoughts, and ARGOS. The performance of Probabilistic Abductive CommonSense is better than other baselines over three benchmarks.

**Compliance With Llm Reviewing Policy:**

Affirmed.

**Final Justification:**

The new diagram has helped better understand the process that they've developed.

**Key Questions For Authors:**

See weaknesses

**Limitations:**

Yes

**Strengths And Weaknesses:**

Strengths:
1) The motivation is sound and well-supported.
2) The approach aims to be impactful and cost-effective
3) Multiple baselines and datasets are used in the experiments.

Weakness:
1) The paper is written in a very abstract way with little to no examples. The lack of concrete examples to illustrate the approach and its need hinders the motivation of the approach and it's usefulness. How is this concretely used in the dataset that you evaluate?
2) The adaptation of the datasets, FOLIO, CosmosQA and QUAIL could be explained with more detailed examples.

---

> ### Author Rebuttal · Authors · 2026-03-31
>
> We are grateful to the reviewer for the insightful and detailed
> commentary and criticism. We are also pleased to see the reviewer's
> appreciation for the well-soundedness of our motivation, the high impact
> our method delivers, and the robust experimental setting.
>
> ## Weaknesses
>
> 1.  *The paper is written in a very abstract way with little to no
>     examples. The lack of concrete examples to illustrate the approach
>     and its need hinders the motivation of the approach and its
>     usefulness. How is this concretely used in the dataset that you
>     evaluate?*\
>     This is a fair criticism, and we agree that a concrete example would
>     help make the paper easier to follow. To address your concern, we
>     will include a new diagram (https://imgur.com/a/6GYq9Ax) that
>     describes how our PACS algorithm would answer a query.
>
> 2.  *The adaptation of the datasets, FOLIO, CosmosQA and QUAIL could be
>     explained with more detailed examples.*\
>     In the original submission, we provided an example of the adaptation
>     of a datapoint from CosmosQA in the original Appendix. This example
>     demonstrates the logical translation and the way it is included in
>     the prompt, as well as the expected output format during LLM
>     generation. It also demonstrates the particular phrasing chosen for
>     the instruction and query. In addition, we provided in the main text
>     an example of the abduction adaptation we conducted for FOLIO at
>     line 257. With that said, the Figure we intend to add to the paper,
>     https://imgur.com/a/6GYq9Ax , provides further understanding of the
>     adaptation of the datasets, and their treatment by our method.

---

> > ### Author Rebuttal · Reviewer_6UFs · 2026-04-04
> >
> > Your new diagram helps a lot understand the process, I would advise using this as a running example through the paper to improve the presentation.

---

> > > ### Author Response · Authors · 2026-04-07
> > >
> > > We thank the reviewer again for the careful discussion and constructive feedback. We will surely take your advice and use the figure and example throughout the paper as an example!

---

### Official Review · Reviewer_NErP · 2026-03-13

**Soundness:** 2
**Presentation:** 2
**Significance:** 2
**Originality:** 2
**Overall Recommendation:** 3
**Confidence:** 3

**Summary:**

This work is about abductive commonsense reasoning that explicitly models probabilistic commonsense.

**Compliance With Llm Reviewing Policy:**

Affirmed.

**Key Questions For Authors:**

1. How does the size and power of the underlying LLM affect the quality of sampled epistemic statements?

**Limitations:**

Yes.

**Strengths And Weaknesses:**

### Strength

1. The mix of probabilistic human consensus modeling with a neurosymbolic search algorithm stands out as fresh. Past works have used LLMs to search for logical premises. But treating the generated propositions explicitly as samples from a latent distribution of individual human epistemic beliefs marks a basic and valuable concept shift.

### Weaknesses

2. For the formulation in Section 4, it will be better to have an simple example.

---

> ### Author Rebuttal · Authors · 2026-03-31
>
> We thank the reviewer for the careful and considerate reading and review
> of our work. We are grateful for the reviewer's recognition of the
> novelty and value of our framing and approach.
>
> ## Weaknesses
>
> 1.  *For the formulation in Section 4, it will be better to have an
>     simple example.*\
>     We thank the reviewer for this suggestion. To address your concern,
>     we will include in the paper a new diagram
>     (https://imgur.com/a/6GYq9Ax) describing how PACS would answer a
>     query for a specific example. We will also include the following
>     paragraph in the main text, at the start of the methodology section,
>     referencing this diagram. "In this section, we will discuss our
>     approach to sampling various distinct reasoners with separate
>     personal beliefs by sequentially selecting and setting individual
>     beliefs. We envision this as a tree search, where at each step we
>     must select from some list of candidate beliefs. Each step in this
>     process can be viewed as the selection of a step during COT. In the
>     Figure below, we demonstrate how this process might unfold,
>     addressing ambiguity and finding multiple viable proof paths. At
>     each step, the score is computed by evaluating # $\sigma_i$ using a
>     SAT solver tool."
>
> ## Questions
>
> 1.  *How does the size and power of the underlying LLM affect the
>     quality of sampled epistemic statements?*\
>     We have re-run the experiment in Table 1 on a newer, larger model:
>     Llama 3.3 70B. We have only had time to test the model on two of the
>     datasets: FOLIO and QUAIL. However, we intend to complete this
>     experiment on all the datsetes. Results show that PACS performs
>     well. On FOLIO, we see that PACS has 88% accuracy. Self-consistency
>     shows 77% accuracy, LoT shows 70% accuracy, ARGOS shows 80% and
>     IF-Beam 78%. On Quail we see that PACS has 84% accuracy, SC has 75%,
>     LOT 72%, ARGOS 80%. For reference, we also tested FOLIO and Quail on
>     GPT-OSS 120B for SC, since the point was raised that this model is
>     trained especially for COT. On this model, SC has 82% on FOLIO and
>     80% on Quail. We note that PACS with Llama 3.3 70B, which is a year
>     older and 50B parameters smaller, outperforms SC on the specially
>     trained thinking model. Additionally, we note that the thinking
>     model performs similarly to PACS on Llama 3 8B.

---

### Decision · Program_Chairs · 2026-04-30

**Decision:**

Accept (regular)

**Comment:**

This work aims to enhance the inference capabilities of large language models (LLMs) through commonsense abductive reasoning and has received four reviews. Three of the reviews are thorough and agree on a score of "weak accept." Additionally, the rebuttal phase was  productive, allowing the authors to effectively address most of the reviewers' concerns. At this point, I see no major reason to reject the paper. If the manuscript is accepted, I strongly encourage the authors to include their responses to the reviewers in the camera-ready version.